# Molecular Response of *Meyerozyma guilliermondii* to Patulin: Transcriptomic-Based Analysis

**DOI:** 10.3390/jof9050538

**Published:** 2023-04-30

**Authors:** Qiya Yang, Xi Zhang, Dhanasekaran Solairaj, Yu Fu, Hongyin Zhang

**Affiliations:** School of Food and Biological Engineering, Jiangsu University, Zhenjiang 212013, China; yangqiya1118@163.com (Q.Y.); xi15612806202@163.com (X.Z.); fuyu900@126.com (Y.F.)

**Keywords:** *Meyerozyma guilliermondii*, patulin, detoxification, molecular response, transcriptome

## Abstract

Patulin (PAT), mainly produced by *Penicillium expansum*, is a potential threat to health. In recent years, PAT removal using antagonistic yeasts has become a hot research topic. *Meyerozyma guilliermondii*, isolated by our group, produced antagonistic effects against the postharvest diseases of pears and could degrade PAT in vivo or in vitro. However, the molecular responses of *M. guilliermondii* over PAT exposure and its detoxification enzymes are not apparent. In this study, transcriptomics is used to unveil the molecular responses of *M. guilliermondii* on PAT exposure and the enzymes involved in PAT degradation. The functional enrichment of differentially expressed genes indicated that the molecular response mainly includes the up-regulated expression of genes related to resistance and drug-resistance, intracellular transport, growth and reproduction, transcription, DNA damage repair, antioxidant stress to avoid cell damage, and PAT detoxification genes such as short-chain dehydrogenase/reductases. This study elucidates the possible molecular responses and PAT detoxification mechanism of *M. guilliermondii*, which could be helpful to further accelerate the commercial application of antagonistic yeast toward mycotoxin decontamination.

## 1. Introduction

*Penicillium expansum* is a toxic secondary metabolite of filamentous fungi that causes serious pollution of foodstuffs, produces substantial economic losses worldwide, and harms consumers’ health [1]. *P. expansum* is the most important producer of PAT, which causes the decay and deterioration of fruits and vegetables. PAT contamination in fruits and their products is conventionally controlled by physical, chemical, and biological strategies. The physical methods mainly control the generation of PAT by means of manual screening, high-pressure washing, refrigeration, etc. However, quality loss, high costs, and environmental pollution are major drawbacks of physical methods [2]. The chemical PAT control methods can be divided into two types; the first type is the prevention of PAT contamination by killing PAT producers through chemical fungicides, and the other type is the removal of PAT by adding chemicals. However, some PAT-controlling chemicals are toxic in nature and can damage the product quality, taste, and nutritional value [2]. The biological methods refer to antagonistic microorganisms (yeasts, bacteria, mold) to control the generation of PAT by inhibiting pathogenic fungal infection or using microorganisms to adsorb or degrade the generated PAT. The biological control methods have attracted more and more researchers’ attention and show a broad application prospect because of their high efficiency, low cost, safety, and non-toxicity [2].

In recent years, the use of antagonistic yeast to remove PAT has become a hot research topic, but the PAT clearance mechanism is unclear. Researchers published two primary PAT scavenging mechanisms of antagonistic yeast; the first is the adsorption of PAT by antagonistic yeast cells without affecting the cell activity. For example, the PAT adsorption capacity of live and heat-inactivated *Saccharomyces cerevisiae* cells is similar, and the yeast cell wall could reduce up to 35.5% PAT in a buffer solution [3]. Another possible PAT removal mechanism of antagonistic yeast is enzymatic degradation during fermentation, which could be induced. For instance, PAT-induced *Rhodosporidium paludigenum* could degrade PAT into desoxypatulinic acid or into E-ascladiol by *Candida guilliermondii*, which occurred within the yeast cells [4,5].

At the beginning of the 21st century, domestic and foreign researchers started exploring the molecular mechanism of PAT degradation by antagonistic yeast. With the advent of “omics” technologies, the studies on PAT biocontrol mechanisms reached the next level. Transcriptome high-throughput sequencing technology has been widely used in various fields to mine molecular research information. Transcriptome refers to collecting all RNAs transcribed by tissues and cells under certain physiological conditions. Almost all the genetic information of proteins comes from the genome, and the regulation of life activities of organisms at the transcriptional level is pervasive and important [6]. Chen et al. found that in *C. guilliermondii*, PAT degradation took place inside the cells, and 30 different proteins involved in 10 biological processes were differentially regulated during the degradation process. Moreover, PAT significantly induced the expression of short-chain dehydrogenase (SDR) at the protein and mRNA levels [5]. A previous study by our research team confirmed the antagonistic effect of *Meyerozyma guilliermondii* (1 × 10^8^ cells/mL) on the postharvest diseases of pears and PAT degradation efficiency in vivo [7]. Previous studies have shown that *M. guilliermondii* can effectively control *P. expansum* in the wound as well as the whole fruit and effectively degrade PAT [8,9]. Still, the PAT removal (detoxification) mechanism of *M. guilliermondii*, especially the molecular responses underlying PAT tolerance, is unclear.

Hence, this study aims to further unveil the molecular responses of *M. guilliermondii* during PAT exposure and detoxification. The transcriptome technology was adopted to study the molecular regulation of *M. guilliermondii* and the gene expression levels on PAT response. Differentially expressed genes (DEGs) were further analyzed to elucidate the PAT stress response mechanisms of *M. guilliermondii* at the molecular level.

## 2. Materials and Methods

### 2.1. Yeast

*M. guilliermondii* was isolated from the pear surfaces from unsprayed orchards by our group and deposited in CCTCC of Wuhan University with the accession number M2017270. In our laboratory, *M. guilliermondii* isolates were maintained at 4 °C in nutrient yeast dextrose agar (NYDA) medium (nutrient broth 8 g, yeast extract 5 g, glucose 10 g, agar 20 g, in 1 L of distilled water). Liquid cultures of the yeast were cultivated in 50 mL of NYD broth (NYDB) by inoculating a loop of *M. guilliermondii* cells and incubating on a rotary shaker at 28 °C for 24 h. Later, the cells were washed twice using sterile distilled water and adjusted to an initial concentration of 1 × 10^8^ cells/mL before experiments. For PAT exposure, 1 mL of *M. guilliermondii* suspension was added into 50 mL NYDB containing PAT (adjusted concentration to 10 μg/mL) and then incubated at 28 °C, 180 rpm for 24 h. *M. guilliermondii* suspension was added into 50 mL NYDB without PAT and was used as a control group. The cells were washed twice with sterile distilled water by centrifugation for 10 min at 4 °C. The yeast cells were frozen by adding liquid nitrogen, and the sample was stored in the −80 °C refrigerator until further use.

### 2.2. Total RNA Extraction from Yeast

The *M. guilliermondii* in NYDB without PAT induction served as the control group, and the *M. guilliermondii* cultured in PAT-supplemented NYDB served as the experimental group. Each group contained 3 biological replicates. After PAT exposure, the *M. guilliermondii* was quickly ground to powder in a pre-cooled mortar with liquid nitrogen. Total RNA was extracted using a column fungal total RNA extraction kit (Sangon Biotech Co., Ltd., Shanghai, China) according to the manufacturer’s instructions. The total RNA concentration and purity were determined using NanoDrop One (Thermo Fisher Scientific, Waltham, MA, USA), and the integrity was determined using Agilent 2100 biochip analyzer (Agilent, Santa Clara, CA, USA).

### 2.3. High Throughput RNA Sequencing and Bioinformatic Analysis

The prepared samples were sent to Nanjing Jisi Huiyuan Biotechnology Co., Ltd., Nanjing, China, for high-throughput RNA sequencing. The genome of *M. guilliermondii* ATCC 6260 (https://www.ncbi.nlm.nih.gov/Taxonomy/Browser/wwwtax.cgi?mode=Info&id=294746&lvl=3&lin=f&keep=1&srchmode=1&unlock (accessed on 26 April 2023)) from the NCBI database served as a reference genome.

The method of FPKM was used to evaluate the expression levels of genes, and the software of DESeq2 1.40.0 was used to screen differentially expressed genes. The genes with a parameter of false discovery rate (FDR) below 0.05 and absolute fold change greater than or equal to 2 or greater than or equal to 1 were considered differentially expressed genes. Additionally, the method of Gene Ontology (GO) enrichment analysis started with all DEGs being mapped to GO terms in the GO database, then gene numbers were calculated for every term, and significantly enriched GO terms in DEGs comparing to the genome background were defined using the hypergeometric test. The p-value was calculated, and then the p-value went through FDR correction, taking FDR ≤ 0.05 as a threshold. GO terms meeting this condition were defined as significantly enriched GO terms in DEGs. Additionally, the method of the Kyoto Encyclopedia of Genes and Genomes (KEGG) pathway enrichment is the same as the GO enrichment analysis.

### 2.4. Validation of DEGs by RT-qPCR

Based on RNA-seq analysis, 12 DEGs that play a key role in regulating PAT response were selected (Table 1). In this study, a β-tubulin gene of *M. guilliermondii* was used as the reference gene. Primers of selected genes were designed using Primer 6.0 and were purchased from Seigon Biotechnology, Shanghai, China. The primers used in RT-qPCR analysis are shown in Appendix A. RNA samples from *M. guilliermondii* cultures exposed with or without PAT were extracted for RT-qPCR analysis as described above. HiFiScript gDNA Removal RT Master Mix (CoWin Biosciences, Beijing, China) was used to synthesize cDNA by reverse transcription. Real-time PCR amplification was performed using a two-step method and was measured in the Applied Biosystems 7300 Real-Time PCR system (USA). The relative expression levels were calculated using 2^−ΔΔCT^ method. The whole experiment was repeated thrice.

## 3. Results

### 3.1. Sample Relationship Analysis

The quality control of the transcriptomic data of all the samples (CK-1, CK-2, CK-3, PAT-1, PAT-2, and PAT-3) showed a quality score above Q30, indicating high sequencing quality and good reliability (Figure 1a). The normalized mapping rate of the transcriptomic data of all the samples with the reference genome reached more than 96%, and the samples were mapped predominantly to the exonic regions (more than 84%). These indicators specified that the transcriptomic data of all the samples were of good quality, and the alignment with the selected reference genome was complete (Appendix A).

The Pearson correlation coefficients between the expression levels of two randomly selected samples were calculated and visually displayed in heat maps to show the correlation between the samples. The R-value was positively correlated, and a larger R-value indicates a better correlation. The R-value of two freely chosen control group samples was greater than 0.99, and the R-value of two freely chosen experimental groups was greater than 0.95, while the R-value of two freely chosen control and experimental group samples was less than 0.9, which showed that the parallelism of the treatment groups was good and the difference between the treatment groups was significant (Figure 1b).

### 3.2. Differential Expression of Genes

When the absolute of log_2_(Fold Change) greater than or equal to 1 and the FDR less than 0.05 were used as the screening criteria, the up-regulated DEGs were 215 and the down-regulated DEGs were 133. When the absolute of log_2_(Fold Change) greater than or equal to 2 and the FDR less than 0.05 were used as the screening criteria, the up-regulated DEGs were 54 and the down-regulated DEGs were 8 (Figure 1c). The DEGs identified in the transcriptome analysis were annotated using NR, Swiss-Port, GO, COG, KOG, KEGG, and other databases (Appendix A).

### 3.3. GO Enrichment Analysis of DEGs

The GO enrichment analysis of the DEGs of the PAT-induced *M. guilliermondii* was enriched in three major categories including biological process, molecular function, and cellular components (Figure 2a). The cell components contained 12 secondary subclasses; among them, membrane and membrane parts might be related to the transport of PAT by *M. guilliermondii* into cells. The molecular function category included 11 secondary subclasses; among them, the enriched transporter activity subclass might be related to the intramolecular transport of PAT, and the catalytic activity and antioxidant activity might be closely related to the stress resistance of *M. guilliermondii*. The biological processes consisted of 20 secondary subclasses; among them, signaling, response to stimulus, biological regulation, and detoxification were related to the growth and development of *M. guilliermondii* and the regulation of PAT and stimulus-response.

Nine secondary subclasses related to the regulation of *M. guilliermondii* response toward PAT were selected for the GO tertiary enrichment analysis (Figure 2b). Among them, peroxisomal membrane, oxidoreductase activity, glutathione transferase activity, drug transmembrane transporter activity, amino acid transmembrane transporter activity, and regulation of fungal-type cell wall organization were related to PAT degradation by *M. guilliermondii*.

### 3.4. COG Enrichment Analysis of Differentially Expressed Genes

The COG enrichment analysis was performed for the DEGs of the PAT-induced *M. guilliermondii* (Figure 3). The top seven functional classifications and enriched DEGs in COG enrichment were amino acid transport and metabolism (42 DEGs); carbohydrate transport and metabolism (38 DEGs); inorganic ion transport and metabolism (29 DEGs); post-translational modification, protein turnover, and chaperones (18 DEGs); energy production and conversion (13 DEGs); lipid transport and metabolism (12 DEGs); and secondary metabolites biosynthesis, transport, and catabolism (12 DEGs).

### 3.5. KEGG Pathway Enrichment Analysis of DEGs

As shown in Figure 4a, all DEGs were annotated to the KEGG database and enriched in 4 primary classifications, such as cellular processes, environmental information processing, genetic information processing, and metabolism, which consisted of 17 secondary subclasses. Among the secondary subclasses, transport and catabolism, signal transduction, replication and repair, amino acid metabolism, carbohydrate metabolism, and metabolism of other amino acids were related to the growth, stress resistance, and PAT-responsive regulation of *M. guilliermondii*. The DEGs of these 6 secondary subclasses were further analyzed via KEGG tertiary enrichment, and the DEGs were enriched in 22 tertiary subclasses (Figure 4b). Four pathways, such as peroxisome pathways, MAPK signaling pathway yeast, glycolysis/gluconeogenesis pathways, and glutathione metabolic pathways, were significantly enriched and correlated with the PAT response regulation in *M. guilliermondii*.

### 3.6. Differential Expression of Genes Involved in PAT Detoxification

As shown in Table 2, the up-regulated expression of ten short-chain dehydrogenase/reductase (SDR) genes and the glutathione S-transferase encoding gene (*gedE*) was observed in PAT-exposed *M. guilliermondii*. Among the SDRs, enoyl-(Acyl carrier protein) reductase, NAD-dependent reductase, and NAD(P) H-dependent reductase family genes were more significantly up-regulated.

### 3.7. Validation of RNA-seq Data by RT-qPCR

Twelve DEGs were randomly selected for RT-qPCR verification to prove the reliability of the transcriptome sequencing results. As shown in Figure 5, the expression levels of these DEGs were analyzed via the regression analysis with the expression levels in the transcriptome data. The expression trends of the selected DEGs were consistent between the transcriptomic analysis and the RT-qPCR analysis; the Pearson correlation coefficient was 0.764, indicating that the transcriptome sequencing results were authentic and reliable.

## 4. Discussion

Pome fruits, including pears, are vulnerable to mechanical damage during cultivation, transportation, storage, and selling, which facilitates pathogen infestations and mycotoxin contamination. PAT is one of the critical mycotoxins that can cause severe ill effects to consumers and economic losses to producers. *P. expansum* is the primary producer of PAT. In recent years, PAT detoxification using antagonistic yeasts has become a hot research topic. Our previous research proved the antagonistic effect and PAT degradation ability of *M. guilliermondii*, but the PAT removal, especially the molecular mechanism, is unclear. In this study, we analyzed the molecular responses of *M. guilliermondii* with or without PAT stimulation through transcriptome sequencing and validated the results via the RT-qPCR analysis. The RT-qPCR results confirmed that the expression trend of the selected DEGs in RT-qPCR and the transcriptomic sequencing results were consistent (Figure 6). The gene annotation analysis of DEGs in GO, COG, KEGG, Swiss-Port, and other databases provided the essential information about the critical genes involved in the molecular regulation of *M. guilliermondii* during PAT exposure. Notably, the genes related to post-translational modification, DNA damage repair, resistance and drug resistance, oxidative stress resistance, growth and reproduction, transcription, and translation regulation and transport (Figure 6) were differentially regulated in *M. guilliermondii*.

The gene *STE14* directs a range of post-translational reactions, including isopentenylation, endoproteolysis, and carboxymethylation [10]. The gene *PPQ1* regulates mating signaling by targeting at or upstream of the terminal MAP kinase Fus3 in the cascade and is also associated with the dephosphorylation of target pathway proteins [11]. In our results, both the *STE14* and *PPQ1* genes related to the protein modification were up-regulated in *M. guilliermondii*. This result indicated that *M. guilliermondii* encounters the PAT induction by up-regulating the genes related to protein modification, ensuring the carboxymethylation and phosphorylation of post-translational proteins and signal processing (Figure 6).

PAT causes ROS accumulation inside the cells, which leads to oxidative DNA damage [12,13]. DNA damage response (DDR) plays a key role in maintaining genome integrity and stability. The protein encoded by *SPBC2A9.02* genetically interacted with the DNA replication initiation proteins Abp1 is a guarantee for the efficient initiation of DNA replication [14]. Likely, the DNA repair protein encoded by *RAD14* is a DNA damage recognition factor in nucleotide excision repair [15]. The DNA repair protein encoded by *MAG1* is associated with a regulatory factor (*RPN4*)-dependent DNA repair pathway. Similarly, the DNA repair factor IIH helicase subunit encoded by *SSL2* [16] is a component of the DNA repair factor and participates in DNA damage repair. In this study, the expression of *SPBC2A9.02*, *RAD14*, *MAG1*, and *SSL2* were all up-regulated, which suggested that the DNA damage caused by PAT stress was repaired by *M. guilliermondii*, which ensured the genome integrity and stability (Figure 6).

A proteomic study of PAT-exposed *C. guilliermondii* indicated that PAT causes adverse stress and affects various metabolic pathways of yeast cells [6]. In the present study, PAT induces adverse stress on *M. guilliermondii* and stimulates the stress and drug resistance mechanisms of the yeast. Some genes associated with stress and drug resistance were up-regulated in *M. guilliermondii*. *APD1* plays a critical role in cellular defense, and the loss of *APD1* leads to the loss of cellular sensitivity and intracellular redox homeostasis [17]. The ubiquitin-binding enzyme E2 encoded by *UBC2* helps the cells adapt to cold, salt, and toxicity stress and improves the ability of environmental stress resistance [18,19]. Likewise, the drug-resistant protein encoded by *MDR1* is associated with the multipotent and pleiotropic drug resistance of yeast [20]. In our research, under PAT stimulation, the expression of *APD1*, *UBC2*, and *MDR1* related to adverse environmental stress and drug resistance was significantly up-regulated in *M. guilliermondii*. The up-regulation of these genes could enhance the resistance of *M. guilliermondii* to PAT and help *M. guilliermondii* to resist PAT toxicity [21] (Figure 6).

After entering the yeast cells, PAT causes sulfhydryl GSH depletion, induces intracellular ROS accumulation, and eventually leads to redox homeostasis imbalance, thus poisoning the cells [6,22]. In *M. guilliermondii*, the oxidative stress-related genes, such as *GRP2* (encoding NADPH dependent methylglyoxal reductase), *NPY1* (encoding peroxisome), and *PST2* (encoding oxidative stress protein) [23,24] were up-regulated to maintain intracellular redox homeostasis. Iron reductase encoded by *CFL1* plays an important role in oxidative stress, and Xu et al. confirmed that mutation in the *CFL1* gene leads to high levels of ROS production in *C. albicans* [25]. The mitochondrial transporter (*FLX1*) encoded by *FLX1* catalyzes the movement of the redox cofactor FAD through the mitochondrial membrane and affects the ATP production efficiency, ROS homeostasis, and longevity of *S. cerevisiae*. The deletion of *FLX1* gene showed significant ATP deficiency and ROS imbalance in *S. cerevisiae* [26]. The peroxisome membrane signal receptor encoded by *PEX5* promotes the peroxisome matrix protein’s introduction by shuttling between the cytoplasm and peroxisome membrane [27]. The genes *GRX2* and *GRX3* encode small redox proteins called glutaredoxins (GRXs), which reduce glutathione as an electron donor and are vital components of the antioxidant system of cells. The loss of GRXs reduces the expression of stress-reactive proteins, resulting in an increased accumulation of ROS in cells [28]. The gene *Snz1* encoding the phosphosynthase subunit is associated with vitamin B6 (VB6) biosynthesis, a potent antioxidant, and plays a vital role in development and stress response [29]. In the present study, the up-regulated expression of *GRP2*, *NPY1*, *PST2*, *CFL1*, *FLX1*, *PEX5*, *GRX2*, *GRX3*, and *Snz1* indicated that *M. guilliermondii* eliminated intracellular ROS and prevented cell damage caused by PAT toxicity by up-regulating the expression of genes related to the redox process (Figure 6).

A series of genes related to growth and reproduction were also up-regulated in PAT-induced *M. guilliermondii*. The gene mel1 is a metabolism-associated gene that encodes α-galactosidase, a key enzyme in the catabolic pathway of galactose and glucose disaccharide [30], and the gene *ETR1* that encodes enoyl reductase is associated with yeast fatty acid synthesis (fatty acid synthesis type II) and respiratory metabolism [31]. The flavodoxin-like genes *YCP4* regulate the expression of several metabolism-related genes during the late growth stage of yeast [32]. The *PKR1* gene encodes the V-type ATPase assembly factor (*PKR1*) associated with iron ion utilization, and *PKR1* deficiency leads to V-ATPase levels and defective Fet3p, a component of the high-affinity iron transport system [33]. The gene *MIA40* encodes the mitochondrial intermembrane space import and assembly protein 40, which uses cytochrome oxidase copper chaperone as an important substrate (encoded by *COX17*) that plays a crucial role in the import, oxidation, and folding of other mitochondrial proteins [34]. Mitochondrial proteins encoded by *NFU1* play an important role in the assembly of mitochondrial Fe-S clusters and intracellular iron homeostasis in yeast [35]. In our research, after PAT stimulation, the expression of *mel1*, *ETR1*, *YCP4*, *PKR1*, *MIA40*, *COX17*, and *NFU1* were all increased to provide enhanced energy utilization, regulated mitochondrial function, promoted respiratory metabolism, and enhanced cell vitality in *M. guilliermondii*, thus reducing cell damage caused by PAT stress (Figure 6). Similarly, in the case of *C. guilliermondii*, PAT stimulation induced the accumulation of heat shock protein 70 to prevent the damage caused by PAT to the yeast cell [6].

In addition to growth-related genes, our transcriptome results also found differential regulation of a range of genes related to transcription and translation. Cyclins (encoded by *Pch1*) form a complex with cyclin-dependent kinase 9 (encoded by *Cdk9*), a forward transcription extension factor, to regulate the capping and elongation of transcripts [36]. The *BUD31* of yeast contributes to spliceosome assembly, thus promoting effective pre-mRNA splicing [37]. The gene *RPC1* encodes the RNA polymerase III subunit, an important part of RNA polymerase III, involved in tRNA, rRNA, and other essential RNA synthesis [38]. *MPP10* encodes the U3 small nucleolar RNA protein (MPP10), an instantaneous correlation factor of eukaryotic ribosomal synthesis, and *MPP10* forms a protein complex in 90S ribosomal precursors, which carries out early processing of 18S rRNA [39]. In our results, *Pch1*, *Cdk9*, *BUD31*, *RPC1*, and *MPP10* were up-regulated in *M. guilliermondii* after PAT stimulation. The up-regulation of these genes could reduce and prevent the transcription and translation blockage caused by PAT to ensure a smooth transcription and translation process (Figure 6).

PAT exposure in yeasts significantly up-regulates the genes that regulate transporters [40]. PAT stimulation in *M. guilliermondii* up-regulated 11 transport-related genes, including the *YOR1* [41], which encodes oligomycin-resistant ATP-dependent permease, and *dotC* [42], which encodes an efflux pump. Both *YOR1* and *dotC* belong to the genes encoding the ATP binding box or major promoter superfamily transporter associated with pumping toxic substances out of cells. Likewise, the nicotinic acid transporter (encoded by *TNA1*) helps the extracellular quinolinic acid to enter cells, thereby increasing intracellular NAD+ concentration [43]. NAD+ biosynthesis is associated with the yeast’s lifespan, and increased intracellular NAD+ levels could prolong the yeast’s lifespan [44]. Remy et al. showed that *TPO1* endows Arabidopsis with multiple drug resistance (MDR). Yeast expressing *TPO1* showed higher tolerance to many herbicides and fungicides. In contrast, yeast mutants lacking *TPO1* showed sensitivity to many drugs [45]. Zn is an important cofactor of transcription factors and enzymes and is essential to all organisms. The gene *zrt1* encodes Zn-regulating membrane proteins that can maintain the dynamic balance of Zn in cells and maintain cell homeostasis [46]. Similarly, the pantothenic acid transporter (encoded by *liz1*) is responsible for the transport of vitamin pantothenic acid and CoA, located in the plasma membrane, which is necessary for cell pantothenic acid uptake [47]. In our research, the expression of *YOR1*, *dotC*, *TNA1*, *TOP1*, and *zrt1* all showed up-regulation, which might be associated with inorganic ions and nutrient utilization. These molecular events could maintain the homeostasis of *M. guilliermondii* cells and enhance cell viability. At the same time, the up-regulation of proteins related to drug transport might be related to the intracellular transport of PAT (Figure 6).

Short-chain dehydrogenase/reductases (SDR) are the largest and most diverse enzyme superfamilies in all life forms, including bacteria, fungi, plants, and animals. SDRs possess broad substrate specificity and multiple biological functions, such as lipid, amino acid, steroid hormone biosynthesis, and xenobiotic metabolism [48]. The role of SDRs in PAT detoxification and induced expression of SDR genes under PAT stress was already reported in several yeasts. The SDRs of *Sporobolomyces* sp. [40], *Candida guilliermondii* [6], *R. mucilaginosa* [49], and *S. cerevisiae* were found to be up-regulated upon PAT stress. Xing et al. (2021) studied the direct involvement of SDRs in PAT detoxification by cloning the *CgSDR* gene from *Candida guilliermondii* into *E. coli.* They found that the purified *CgSDR* protein could reduce 80% of PAT from apple juice. The *CgSDR* could transform toxic PAT into non-toxic E-ascladiol in vitro with NADPH as a coenzyme [50]. In the present study, the significant up-regulation of several SDRs was observed in *M. guilliermondii* under PAT stress, which speculated that SDRs could be involved in PAT detoxification [51]. Similarly, the role of glutathione S-transferase (GST) in PAT detoxification was described from the proteomic analysis of PAT-induced *R. mucilaginosa*. GST could catalyze the conjugation of a reduced form of GSH with PAT in *R. mucilaginosa* upon PAT induction [49]. In our results, the GST expression was induced in *M. guilliermondii* under PAT stress. Collectively, our study supports the previously proposed hypothesis of PAT detoxification into E-ascladiol via SDRs and the involvement of GST. Studies on the direct participation of SDRs and GST of *M. guilliermondii* in PAT detoxification and the degradation products by heterogeneous expression and purification are ongoing.

## 5. Conclusions

In summary, through transcriptome sequencing and the subsequent bioinformatics analysis, this study revealed the changes in the gene expression during the response regulation of *M. guilliermondii* to PAT. Based on the transcriptome results, the genes related to post-translational modification, DNA damage repair, resistance and drug resistance, oxidative stress resistance, growth and reproduction, transcription, and translation regulation and transport were excavated, and the genes involved in PAT detoxification were distinguished. Collectively, the findings presented in this study will serve as a foundation for the further understanding of the molecular responses and the PAT detoxification mechanisms of *M. guilliermondii*. Transcriptome analysis will provide more insights for further research and development of PAT biodetoxification strategies.

## Figures and Tables

**Figure 1 jof-09-00538-f001:**
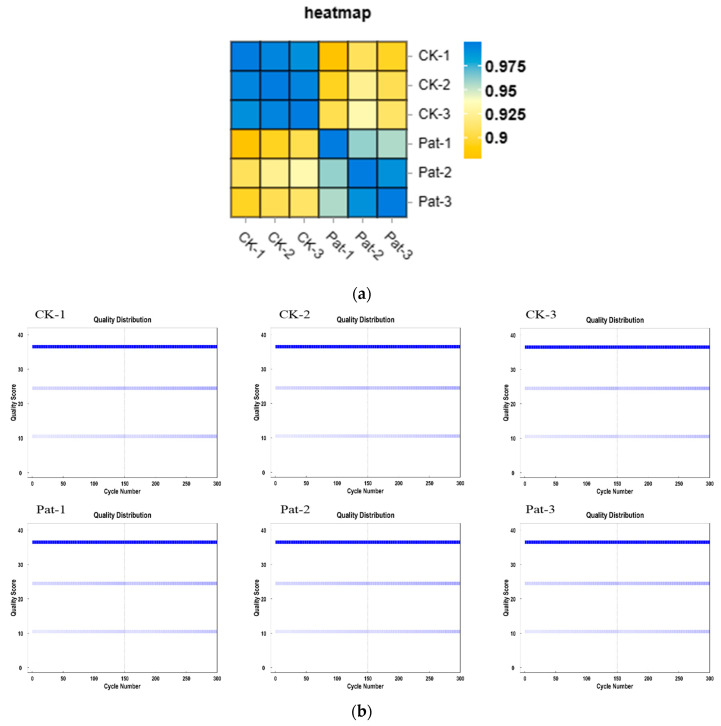
(**a**) Mass distribution map of sequencing base. (**b**) Correlation heat map of samples. CK and Pat corresponds to *M. guilliermondii* treated with and without PAT, respectively. (**c**) Statistics of annotated DEGs between CK vs. PAT-induced *M. guilliermondii* samples.

**Figure 2 jof-09-00538-f002:**
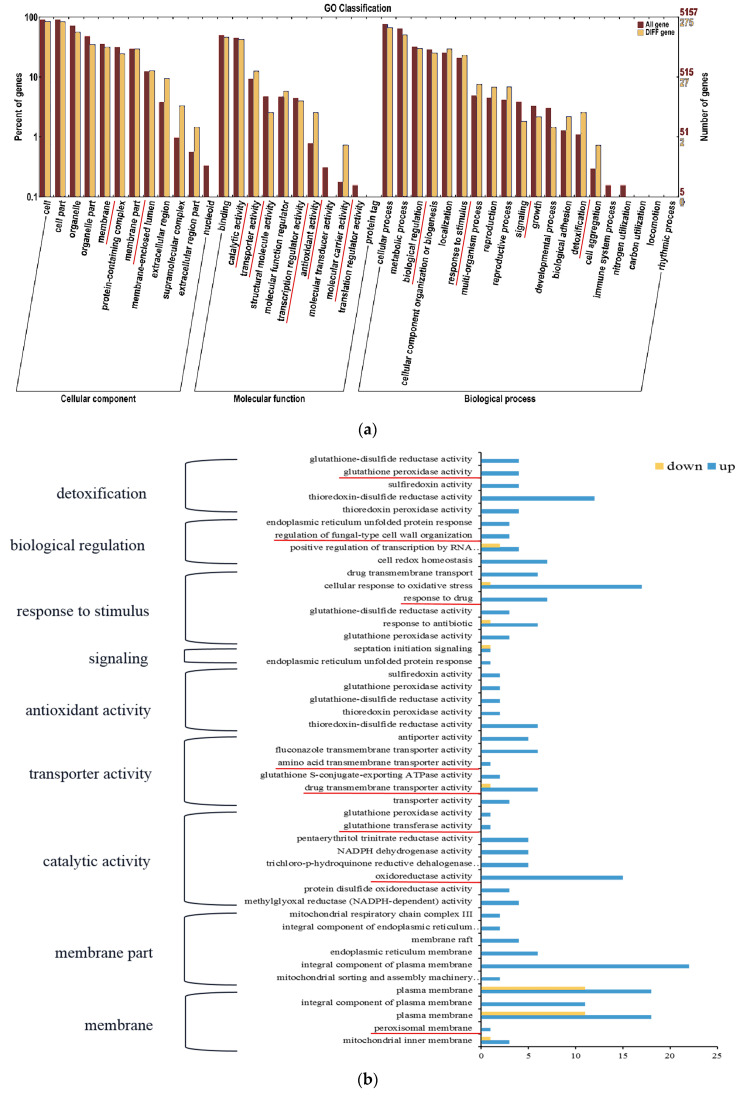
GO enrichment analysis of DEGs. (**a**) GO secondary-level enrichment classification of DEGs (the red lines indicate the subclasses associated with growth and development of *M. guilliermondii*, stimulus response, and regulation of PAT response). (**b**) GO third-level enrichment classification of DEGs (the content marked with red lines is related to the degradation process of PAT response by *M. guilliermondii*).

**Figure 3 jof-09-00538-f003:**
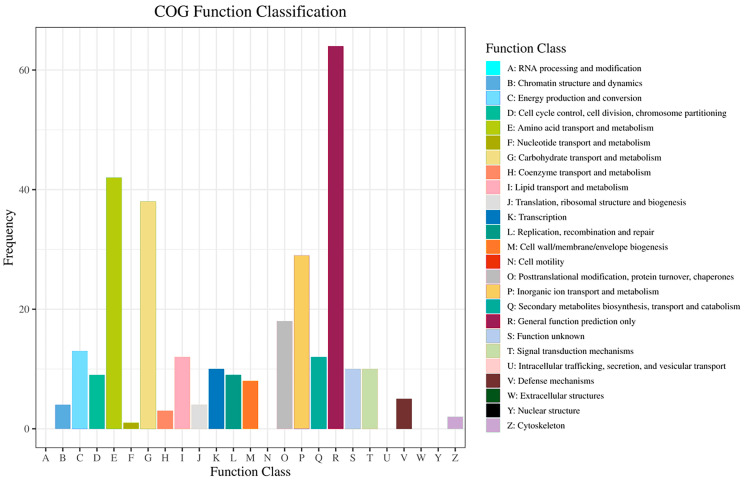
COG enrichment analysis of identified DEGs. The x-axis indicates different classifications, and the y-axis indicates the frequency of DEGs in each COG class.

**Figure 4 jof-09-00538-f004:**
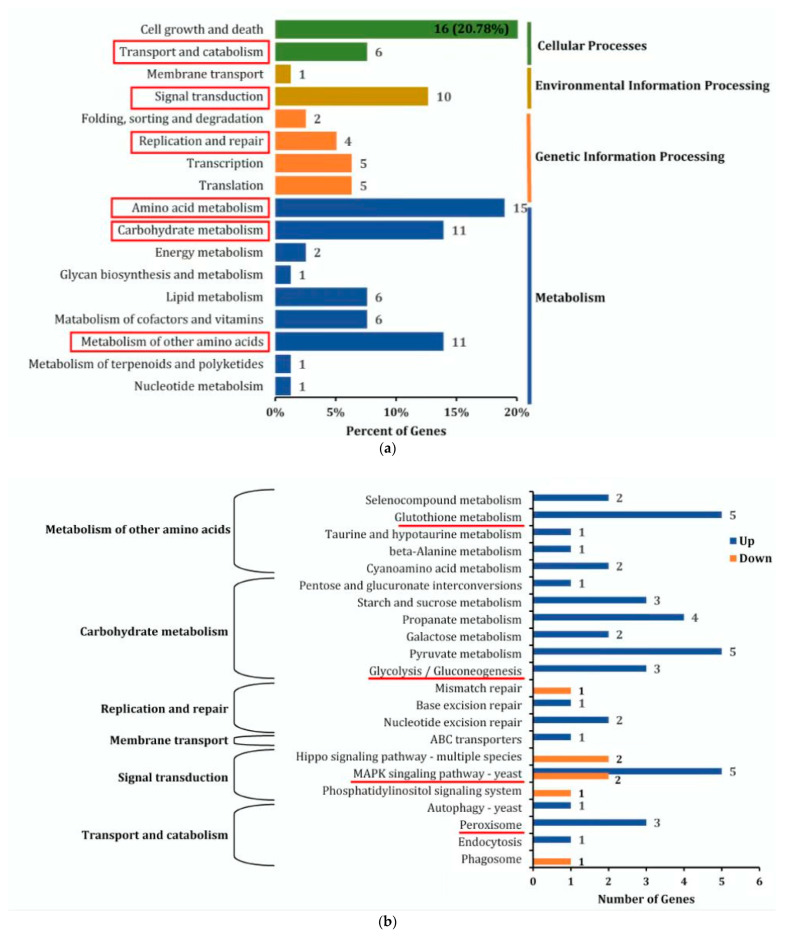
(**a**) KEGG secondary enrichment analysis of DEGs (red boxes are six secondary subclasses related to growth, stress resistance, and response regulation of PAT in *M. guilliermondii*). (**b**) KEGG tertiary enrichment analysis of DEGs (the red lines indicate the highly enriched pathways associated with the regulation of PAT responses).

**Figure 5 jof-09-00538-f005:**
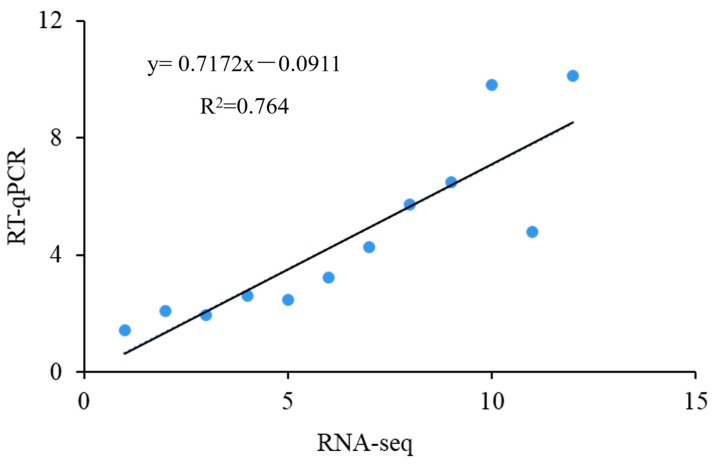
The linear relationship between gene expression values obtained by RT-qPCR and RNA-seq.

**Figure 6 jof-09-00538-f006:**
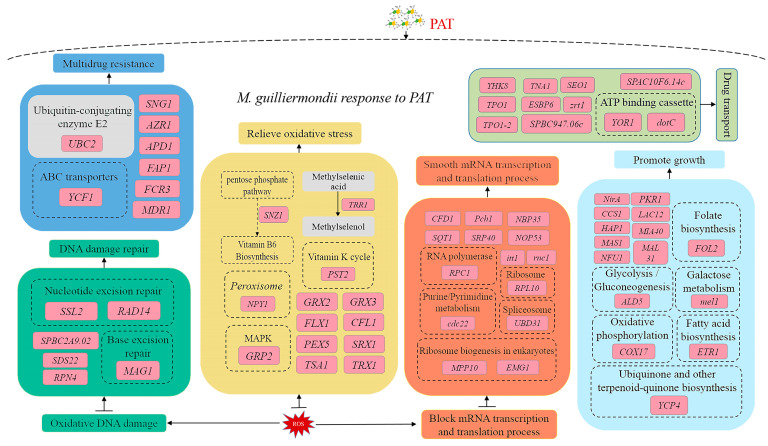
Schematic illustration of proposed mechanisms involved in the molecular responses of *M. guilliermondii* to PAT exposure based on transcriptome.

**Table 1 jof-09-00538-t001:** The genes targeted for RT-qPCR.

Gene ID	Gene Name	Description
PGUG_00994	*NOP53*	Ribosome biogenesis protein
PGUG_01741	*MRR1*	Hypothetical protein
PGUG_01881	*SPCC24B 10.20*	Hypothetical protein
PGUG_05287	*YML 131W*	Hypothetical protein
PGUG_01077	*SPCC 663.08c*	Hypothetical protein
PGUG_04552	*GRP 2*	NADPH-dependent methylglyoxal reductase
PGUG_05192	*YPR022C*	Hypothetical protein
PGUG_04009	*YJR096W*	Hypothetical protein
PGUG_05193	*SPCC663.09c*	Hypothetical protein
PGUG_01005	*GSH 2*	Hypothetical protein
PGUG_03271	*MDR 1*	Multidrug resistance protein 1
PGUG_00888	*YHK8*	Probable drug/proton antiporter

**Table 2 jof-09-00538-t002:** Statistics of key DEGs.

Gene ID	Gene Name	Log_2_ FC	Description
Protein modification			
PGUG_02927	*STE14*	1.05	Protein-S-isoprenylcysteine O-methyltransferase
PGUG_00221	*PPQ1*	1.32	Serine/threonine-protein phosphatase
DNA damage repair			
PGUG_04669	*RAD14*	2.26	DNA repair protein
PGUG_03571 PGUG_00484	*SDS22*	1.91 2.30	Protein phosphatase 1 regulatory subunit
PGUG_04889	*SPBC2A9.02*	1.34	Hypothetical protein
PGUG_01549	*MAG1*	2.65	DNA-3-methyladenine glycosylase
PGUG_04806	*RPN4*	1.14	Transcriptional regulator
PGUG_05450	*SSL2*	1.12	DNA repair factor IIH helicase subunit XPB
Resistance and drug resistance			
PGUG_04879	*YCF1*	1.67	Metal resistance protein
PGUG_03453	*APD1*	1.32	Actin patches distal protein 1
PGUG_05641	*UBC2*	1.23	Ubiquitin-conjugating enzyme E2
PGUG_01353 PGUG_01354 PGUG_00697	*SNG1*	1.80 1.80 3.06	Nitrosoguanidine resistance protein
PGUG_03271 PGUG_05048 PGUG_04886	*MDR1*	10.11 9.43 3.22	Multidrug resistance protein 1
PGUG_03388	*FCR3*	1.67	Fluconazole resistance protein 3
PGUG_03871	*FAP1*	2.08	FKBP12-associated protein 1
PGUG_04884 PGUG_01265	*AZR1*	2.15 2.88	Azole resistance protein 1
Antioxidant stress			
PGUG_03261 PGUG_05692 PGUG_04552 PGUG_05714	*GRP2*	6.70 1.03 4.79 3.40	NADPH-dependent methylglyoxal reductase
PGUG_05342	*CFL1*	1.23	Ferric reductase transmembrane component
PGUG_02968 PGUG_05160	*TRX1*	2.06 1.36	Thioredoxin-1
PGUG_05497	*PEX5*	2.23	Peroxisomal targeting signal receptor
PGUG_04222	*PST2*	1.49	Protoplast secreted protein 2
MSTRG.3296	*GRX2*	1.72	Glutaredoxin-2, mitochondrial
PGUG_03196	*GRX3*	1.11	Monothiol glutaredoxin-3
PGUG_02781	*TSA1*	1.29	Peroxiredoxin TSA1-A
PGUG_02681	*TRR1*	1.25	Thioredoxin reductase
PGUG_03985	*SNZ1*	1.03	Pyridoxal 5 and apos-phosphate synthase subunit
PGUG_05341	*SRX1*	1.87	Sulfiredoxin
PGUG_05768	*NPY1*	1.57	NADH pyrophosphatase
PGUG_00152	*FLX1*	1.21	Mitochondrial FAD carrier protein
Cell wall and membrane formation			
PGUG_04330	*YMR244W*	2.62	Beta-glucosidase (SUN family)
PGUG_03333	*HAP1*	1.13	Heme-responsive zinc finger transcription factor
PGUG_00027	*RBE1*	1.69	Repressed by EFG1 protein 1
Growth and reproduction			
PGUG_04612	*MAL31*	1.32	Maltose permease
PGUG_03041	*FOL2*	1.04	GTP cyclohydrolase
PGUG_03333	*HAP1*	1.13	Heme-responsive zinc finger transcription factor
PGUG_04594	*mel1*	1.81	Alpha-galactosidase
PGUG_01862	*ETR1*	3.37	Enoyl-[acyl-carrier-protein] reductase 1, mitochondrial
PGUG_04221	*YCP4*	1.10	Flavoprotein-like protein
PGUG_00387	*LAC12*	1.69	Lactose permease
PGUG_04842	*nirA*	2.02	Nitrogen assimilation transcription factor
PGUG_03008	*ALD5*	1.03	Aldehyde dehydrogenase 5, mitochondrial
PGUG_00303	*PKR1*	1.03	V-type ATPase assembly factor
PGUG_00474	*CCS1*	1.57	Superoxide dismutase 1 copper chaperone
PGUG_04752	*MIA40*	1.05	Mitochondrial intermembrane space import and assembly protein 40
PGUG_02696	*NFU1*	1.21	NifU-like protein, mitochondrial
PGUG_04929	*COX17*	1.24	Cytochrome c oxidase copper chaperone
PGUG_04481	*MAS1*	1.76	Mitochondrial-processing peptidase subunit beta
Transcription and translation regulation			
PGUG_00994	NOP53	1.41	Ribosome biogenesis protein
PGUG_01029	*NBP35*	1.002	Cytosolic Fe-S cluster assembly factor
PGUG_02131	*CFD1*	1.80	Cytosolic Fe-S cluster assembly factor
PGUG_01264	*Pch1*	2.39	Cyclin pch1
PGUG_01171	*RPL10*	1.45	60S ribosomal protein
PGUG_03610	*BUD31*	1.52	Pre-mRNA-splicing factor
PGUG_04342	*RPC1*	1.50	DNA-directed RNA polymerase III subunit
PGUG_00512	*SQT1*	1.48	Ribosome assembly protein
PGUG_04094	*cdc22*	1.21	Ribonucleoside-diphosphate reductase large chain
PGUG_04921	*MPP10*	1.31	U3 small nucleolar RNA-associated protein
PGUG_03426	*rnc1*	1.13	RNA-binding protein
PGUG_02152	*itt1*	1.01	E3 ubiquitin-protein ligase
PGUG_04056	*SRP40*	3.12	Suppressor protein
PGUG_03307	*EMG1*	1.19	Ribosomal RNA small subunit methyltransferase
Transporter			
PGUG_01272	*SPAC10F6.14c*	1.26	ABC1 family protein
PGUG_03561 PGUG_02559 PGUG_01605 PGUG_03562	*YOR1*	2.20 1.03 1.73 2.22	Oligomycin resistance ATP-dependent permease
PGUG_00888	*YHK8*	1.96	Probable drug/proton antiporter
PGUG_05418	*ESBP6*	2.70	Uncharacterized transporter
PGUG_05416	*TNA1*	1.02	High-affinity nicotinic acid transporter
PGUG_05858	*SEO1*	1.33	Probable transporter
PGUG_03366	*SPBC947.06c*	1.67	MFS-type transporter
PGUG_01124	*TPO1*	3.56	Polyamine transporter 1
PGUG_04286	*TPO1_2*	1.01	Multidrug transporter
PGUG_00124	*dotC*	1.08	Efflux pump
MSTRG.1806	*zrt1*	1.64	Zinc-regulated transporter
PGUG_05857	*liz1*	1.39	Pantothenate transporter

## Data Availability

The transcriptome data of *M. guilliermondii* cultured with and without PAT used in this study were deposited in the NCBI Sequence Read Archive (SRA) Sequence Database with accession numbers SRR24233557 to SRR24233562.

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
