# Peer review of "Molecular Response of Meyerozyma guilliermondii to Patulin: Transcriptomic-Based Analysis"

_jof, 2023, doi:10.3390/jof9050538_

Round 1

Reviewer 1 Report

This work deals with a hot topic in the context of the fruit postharvest diseases, particularly the role of antagonistic yeasts in the Patulin (PAT) removal. The present work describes the molecular response of M. guilliermondii on PAT exposure and the enzymes involved in PAT degradation, using a transcriptomics approach. This method allowed to verify the up-regulation of a great variety of genes involved in several stress-response pathways. This study could be helpful to further studies on antagonistic yeast toward mycotoxin decontamination. The experiments are well described, and they demonstrate robustness and simplicity for application and a base from which start for the next studies.

Minor comments:

-          I suggest checking in the whole text to correct the microgranism names in Italic form

-          Line 109: I suggest modifying the sentence.

-          Line 118: I suggest modifying “2-ΔΔCT” in “2-ΔΔCT”.

-          Line 138-140: I suggest modifying the sentences.

-          I suggest improving the quality of the figure, increasing the font of the writings.

-          I suggest checking in the whole text to correct the name of genes in Italic form

-          I recommend performing an English fine/minor spell check of the text.

Moderate editing of English language

Reviewer 2 Report

The authors performed a transcriptomic study to identify differential gene expression between Meyerozyma guilliermondii (teleomorph of Candida guilliermondii) exposed to patulin in a nutrient medium vs the same yeast in a water control. The design of the control to just be an equivalent volume of water instead of the nutrient medium without added patulin (lines 84-87) introduces a confounding variable, making the results less applicable to the purpose of the study. The RNA-Seq data should be deposited in an open databank like DDBJ or NCBI SRA before submission of the manuscript and the relevant accession numbers must be in the main text. More discussion of the alignment of transcriptomic and proteomic results from Candida patulin degraders is needed, particularly how the current results compare to those from doi: 10.3390/toxins9020048. The authors cite general results from this article’s abstract, but the article is a proteomic equivalent of the current study and merits much more attention from the current authors. There are also numerous typos that need to be fixed, which may or may not be detected by an automated spelling and grammar checker. The suggestions and comments below are not exhaustive. Overall, major revisions are needed before the manuscript should be considered for publication.

Many lines- italicize species names

Lines 14-21- can make sentences easier to understand by combining phrases. For example, instead of upregulated A; upregulated B; upregulated C -> upregulated A, B and C

The authors should add more discussion about M. guilliermondii and how it compares to other known patulin biocontrol agents, previous studies by themselves could be cited: https://doi.org/10.1016/j.biocontrol.2022.104856, https://doi.org/10.1016/j.biocontrol.2021.104692, https://doi.org/10.1016/j.foodchem.2023.135785

Line 86- did the authors mean “or” instead of “and”?

Lines 103-107- read processing methods should be specified, including the software and parameters used

Names of the genes targeted for qRT-PCR should be in the main text along with the methods used. Please also specify correlations paired to the specific genes in a table somewhere

Please use colorblind-friendly palettes in your figures (https://davidmathlogic.com/colorblind; https://towardsdatascience.com/is-your-color-palette-stopping-you-from-reaching-your-goals-bf3b32d2ac49).

Line 122- “transcriptome” -> “transcriptomic”

Line 138-140- it’s better to just state “absolute” rather than use ||

Figure legends do not provide enough explanation and context.

Figures 2 and 4- need to explain why some phrases have red underlines/boxes

Figure 2B- some GOs repeated in x-axis

Figure 5 and its discussion does not belong in the Results section as these are not pathways that were discovered or verified in the current study; pathways should also be cited, but the reviewer suggests that these paragraphs are just removed

Table S1- “reverse” not “revers”

The main text should list the most highly affected DEGs by fold change

Line 364- did the authors mean “flavodoxin” not “flavin toxoid”?

Most of the writing is understandable, but overall could be significantly improved

Round 2

Reviewer 2 Report

Thank you to the authors for the responses to the suggestions. With the correction to the methodology and public release of the raw data, the manuscript is acceptable to proceed to the next stage.

Minor editing may be needed (ex. line 112, change>2 -> change >2 ), but does not significantly deter from overall understanding.
